# Enhancing Sustainability through Ecosystem Services Evaluation: A Case Study of the Mulberry-Dyke and Fish-Pond System in Digang Village

**Shuyang Tang, Ziwei Liu, Yumei Li and Mingqin Zhou ***

Collage of Horticultural and Gardening, Yangtze University, Jingzhou 434025, China;
2021720862@yangtzeu.edu.cn (S.T.); 2021720865@yangtzeu.edu.cn (Z.L.); 2021720872@yangtzeu.edu.cn (Y.L.)
* Correspondence: zhoumqzds@126.com

**Abstract:** The Mulberry-Dyke and Fish-Pond System, as a paradigm of traditional Chinese agricultural recycling models, represents a distinct ecosystem. This study focuses on the Mulberry-Dyke and Fish-Pond System in Digang Village, Huzhou, as a typical case. The village serves as a core conservation base for the Huzhou Mulberry-Dyke and Fish-Pond System, hosting the Huzhou Agricultural Science and Technology Development Center's Academician and Expert Workstation and the world's only Mulberry-Dyke and Fish-Pond System Visitor Center. These facilities provide strategic guidance for the conservation, development, planning, and inheritance of the Huzhou Mulberry-Dyke and Fish-Pond System. Considering the unique environment and limitations in data acquisition, this study employed the Analytic Hierarchy Process (AHP) and Fuzzy Comprehensive Evaluation (FCE) to develop an ecosystem service assessment framework encompassing eight aspects and 29 factors assessing the provisioning, regulating, and cultural services of the Mulberry-Dyke and Fish-Pond System. The results indicate that the ecosystem services of the Digang Village Mulberry-Dyke and Fish-Pond System perform at a high level, with cultural services playing a significant role in the overall ecosystem services. The regulating services are relatively weak, highlighting deficiencies in mulberry land management, while the capacity of provisioning services is strong. These findings are crucial for understanding the value of ecosystem services in Digang Village's Mulberry-Dyke and Fish-Pond System, identifying management shortcomings, and providing direction for future assessments and management. This study also offers a practical and effective assessment method for ecosystem service evaluation at smaller scales, where the targeted approach and the presence of significant ambiguity and uncertainty in data are prominent.

**Keywords:** Digang Village; analytical hierarchy process; fuzzy comprehensive evaluation; Mulberry-Dyke and Fish-Pond System; ecosystem service evaluation



## 1. Introduction

The Mulberry-Dyke and Fish-Pond System represents a distinctive agroecosystem in China renowned globally for its pond mud fertilizing mulberry, mulberry leaves nurturing silkworms, silkworm sand feeding fish, and fish manure fertilizing mud—the essence of a highly productive agricultural model [1]. With the advancement of technology, agricultural mechanization and facility-based practices have become the dominant trend in the majority of current agricultural models [2,3]. This trend has led to highly efficient agricultural production and larger-scale farming operations, making significant contributions to grain production and the development of global supply chains [4]. However, while modern agricultural models emphasize economic efficiency, they often involve the large-scale consumption of energy and resources, posing potential threats to the environment. Therefore, modern agricultural models face severe challenges in terms of sustainable development [5,6]. In contrast, traditional circular agriculture not only prioritizes economic

gains but also considers ecological preservation and local cultural heritage [7]. The sustainability and ecological principles inherent in this traditional model offer valuable insights for steering the modern agricultural model toward sustainable development.

Ecosystem Services (ES) refer to the advantages that humans obtain from ecosystems, ultimately striving for sustainable human well-being [8,9]. As outlined by the United Nations Millennium Ecosystem Assessment (MA), these services are classified into Provisioning Ecosystem Services (PESs), Regulating Ecosystem Services (RESs), Cultural Services (CESs), and Supporting Ecosystem Services (SESs) [10]. The Mulberry-Dyke and Fish-Pond System, as a representative of Chinese agricultural culture, holds significant value across diverse domains, including agricultural production, environment regulation, and cultural heritage preservation. Evaluating the ecosystem services of the Mulberry-Dyke and Fish-Pond System provides insights into their comprehensive benefits, encompassing food production, ecosystem regulation, and cultural heritage contributions. This assessment aids management and decision-makers in formulating more scientific and rational agricultural strategies, fostering economically, socially, and environmentally sustainable agricultural development.

Currently, methods for assessing ecosystem services are broadly categorized into two types: valuation-based and material-based assessments [11–13]. Valuation-based assessments primarily focus on providing decision-makers with management strategies and grounds for evaluating the market values of services, whereas material-based assessments concentrate on investigating the mechanisms behind the formation of ecosystem services. In existing research on the ecosystem services of the Mulberry-Dyke and Fish-Pond System, scholars have employed these two assessment methods. For instance, using the contingent valuation method, a study assessed the value of nine ecosystem services of the Mulberry-Dyke and Fish-Pond System, revealing that its overall service value exceeds that of separate mulberry gardens and fish ponds combined [14]. Additionally, the emergy (embodied energy) method was used to compare different agricultural ecological engineering models of the system [15], production models from various periods [16], and the differences between the dyke-pond system and traditional agriculture [17], thus evaluating the sustainability of the Mulberry-Dyke and Fish-Pond System model. In research specifically focusing on the Mulberry-Dyke and Fish-Pond System, most of the literature comes from contributions by Chinese scholars, including investigations into the historical eco-economic context of the Mulberry-Dyke and Fish-Pond System [18,19], studies on ecological restoration strategies [20,21], analyses of landscape patterns [22,23], material cycling models [24], and evaluations of sustainability capabilities [25]. However, these studies have not yet comprehensively considered the combined benefits of the services provided by the Mulberry-Dyke and Fish-Pond System, especially in terms of the spiritual well-being obtained by humans, thus necessitating further in-depth research to unveil their comprehensive performance in these areas.

This study focuses on Digang Village within the core conservation area of the Mulberry-Dyke and Fish-Pond System in Huzhou, Zhejiang Province, China. The village, characterized by the typical features of the Jiangnan water town plains, is surrounded by water on all sides and has developed over thousands of years relying on the Mulberry-Dyke and Fish-Pond System. This study adopted a combined approach of the Analytic Hierarchy Process (AHP) and Fuzzy Comprehensive Evaluation (FCE). AHP provides a clear path for breaking down an abstract problem into concrete analytical units, while FCE allows for the establishment of a series of grading standards upon this foundation, facilitating the collection of data from diverse sources. This approach enables the quantitative integration and analysis of information from different dimensions [26,27]. The innovation of this article lies in applying these well-developed methods in combination to assess ecosystem services, offering a flexible strategy to tackle the challenges we face. This method was applied in the Mulberry-Dyke and Fish-Pond System to address the research challenges of the ambiguity of data sources and high uncertainty. Regarding another aspect, MA and other research treat supporting services as ecological functions or ecological processes, such as biomass production, oxygen production, and the water cycle [28–32]. Incorporating supporting services into calculations may lead to redundancy in the assessments. This perspective

resonates with some scholars' skepticism about Costanza's calculation of intermediate and final services leading to redundant estimations [33]. Therefore, we chose provisioning, regulating, and cultural services as the framework of our assessment research.

## 2. Materials and Methods

### 2.1. Study Area

Currently, China has 19 Globally Important Agricultural Heritage Systems (GIAHSs). The Mulberry-Dyke and Fish-Pond System in Huzhou, Zhejiang (located at 37°12′18″ N, 120°17′40″ E), is situated on the plains south of Lake Taihu, within a subtropical climate zone. The annual average temperature ranges from 17.8 °C to 18.2 °C, with annual precipitation between 1348 mm and 1723 mm. In 2017, it was recognized as a GIAHS [34]. Digang Village, located in the Nanxun town of Huzhou, serves as an important monitoring site for the Mulberry-Dyke and Fish-Pond Heritage System. It is home to the world's only Mulberry-Dyke and Fish-Pond Visitor Center and the Huzhou Agricultural Science and Technology Development Center's Academician and Expert Workstation, which focuses on the conservation and planning of the Mulberry-Dyke and Fish-Pond System. The total area of the village is about 643 hectares, consisting of arable land, gardens, forests, grasslands, water bodies, water conservancy facilities, construction land, and other types of land. The Mulberry-Dyke and Fish-Pond System itself covers approximately 220 hm$^2$, with the central village area covering about 130 hm$^2$. The main agricultural activities include freshwater fish farming and sericulture. The most important agricultural species within the system include 2 types of silkworms, 4 types of mulberry trees, 7 types of fish, and 3 types of fruits and vegetables. The village has a registered population of about 1141 households, of which approximately 280 provide heritage tourism services, and the per capita income of farmers operating agricultural heritage is CNY 6500. The village houses the core conservation area of the Huzhou Mulberry-Dyke and Fish-Pond System, covering about 66 hm$^2$, including approximately 23–25 hm$^2$ of mulberry gardens and 33–35 hm$^2$ of fish ponds. This study focuses on the central village of Digang and the core conservation area within the village (Figure 1).

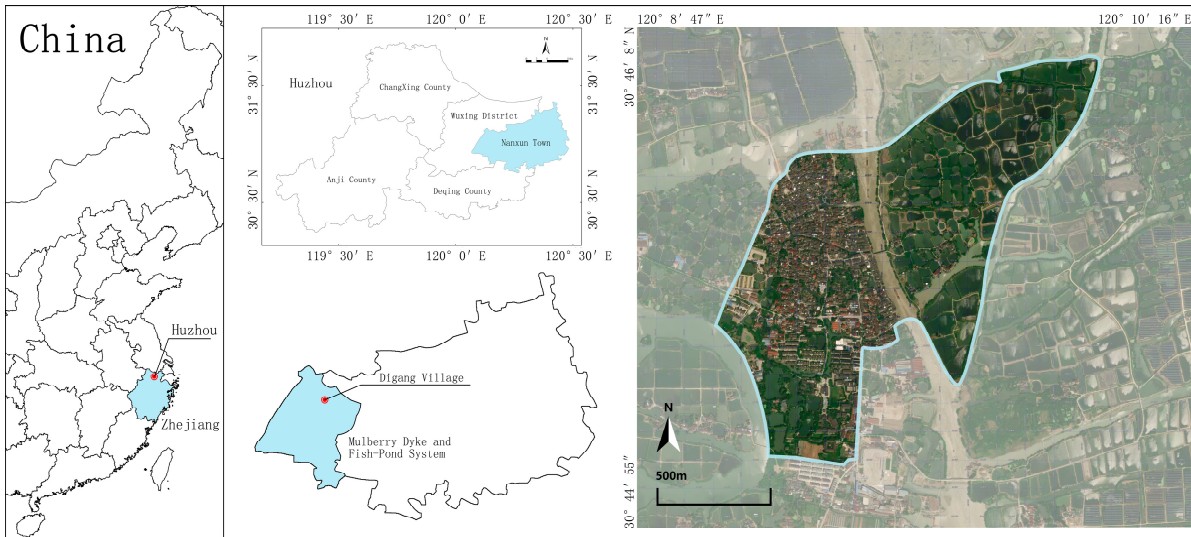

**Figure 1.** Location of the core conservation area of the Mulberry-Dyke and Fish-Pond System in Digang Village, Huzhou, China.

### 2.2. Methodology and Data

#### 2.2.1. Modeling the Valuation of ES

A scientific evaluation index system serves as the prerequisite and foundation for the effective evaluation of ecosystem services in the Mulberry-Dyke and Fish-Pond System [35]. In the classification of ES by the MA [10], Provisioning Ecosystem Services (PESs) encom-

pass services produced or provided by ecosystems, such as food, fiber, genetic resources, etc. RESs are benefits derived from the regulating function of ecosystem processes, including the regulation of atmospheric quality, climate, and the environment, etc. CESs refer to non-material benefits obtained from ecosystems. Additionally, supporting services represent a function of ecosystems necessary for the provision of other services. In the context of the Mulberry-Dyke and Fish-Pond System, PESs, RESs, and CESs were specifically selected to construct a comprehensive ecosystem service evaluation system.

Building upon the existing literature [36–42], the results of discussions with 4 experts in ecology, and an assessment of the operational status of the Mulberry-Dyke and Fish-Pond System, the evaluation index system was designed. Given the significance of mulberry harvesting and fish farming densities in these fishponds [43–45], the production of mulberries and fish was chosen to characterize the PESs. Differences in temperature and humidity between the Mulberry-Dyke and Fish-Pond System and downtown Huzhou City as well as the air quality index of the Mulberry-Dyke and Fish-Pond System were used to characterize the climate regulation value of the RESs, and its basal environmental regulation capacity was flanked by pesticide and fertilizer use in the Mulberry-Dyke and Fish-Pond System, thus characterizing the basal environmental regulation value of the RESs [46–49]. CESs were characterized by 4 aspects: aesthetics, education, leisure and entertainment, and cultural heritage [50,51]. The finalized evaluation index system comprised 1 objective, 3 guidelines, 8 indicators, and 29 factors. The specific framework of the index system is detailed in Table 1 below.

**Table 1.** Ecosystem Service Assessment of the Mulberry-Dyke and Fish-Pond System.

| Target Layer | Criterion Layer | Index Layer | Factor Layer |
|---|---|---|---|
| A Ecosystem service assessment of the Mulberry-Dyke and Fish-Pond System | B1 Provisioning ecosystem services | C1 Mulberry land production value | D1 Mulberry leaf production |
| | | | D2 Mulberry fruit production |
| | | C2 Fishpond production value | D3 Conventional fish farming production |
| | | | D4 Ecological fish farming production |
| | B2 Regulating ecosystem services | C3 Basal environment regulation value | D5 Fertilizer application intensity |
| | | | D6 Pesticide application intensity |
| | | C4 Climate regulation value | D7 Relative humidity adjustment range |
| | | | D8 Average temperature regulation |
| | | | D9 Air quality index |
| | B3 Culture ecosystem services | C5 Aesthetics value | D10 Plant landscape richness |
| | | | D11 Seasonal changes in the landscape |
| | | | D12 Overall harmony |
| | | | D13 Water clarity |
| | | | D14 Leveling and hardening of road surface |
| | | C6 Education value | D15 Fish and mulberry culture education |
| | | | D16 Humanistic tradition |
| | | | D17 Cultural propaganda and exhibition |
| | | | D18 Religious culture |
| | | C7 Leisure and entertainment value | D19 Location conditions |
| | | | D20 Sanitary conditions |
| | | | D21 Tourism infrastructure |
| | | | D22 Agricultural diversity experience |
| | | | D23 Experience of the diversity of tourism products |
| | | | D24 Visual and psychological perception |

**Table 1.** *Cont.*

| Target Layer | Criterion Layer | Index Layer | Factor Layer |
|---|---|---|---|
| A<br>Ecosystem service assessment of the Mulberry-Dyke and Fish-Pond System | B3<br>Culture ecosystem services | C8<br>Cultural heritage value | D25 Village architectural style |
| | | | D26 Village traditional customs |
| | | | D27 Ancient bridges and other historical and cultural features |
| | | | D28 Food culture characteristics |
| | | | D29 Fish and mulberry culture characteristics |

2.2.2. AHP-FCE-Based Ecosystem Service Evaluation Model for the Mulberry-Dyke and Fish-Pond System

The evaluation model consisted of two main parts. First, the Analytical Hierarchy Process (AHP) method was employed to establish the weights for the indicators within the ecosystem service evaluation system for the Mulberry-Dyke and Fish-Pond System. Then, the multilevel Fuzzy Comprehensive Evaluation (FCE) method was applied to conduct a comprehensive assessment of the ecosystem services of the Mulberry-Dyke and Fish-Pond System. The integrated AHP-FCE evaluation model was outlined as follows:

(1)　Establishment of evaluation factor domain and evaluation criteria

Establishment of the evaluation factor domain V for ecosystem services in the Mulberry-Dyke and Fish-Pond Systems, divided into 5 levels (Table 2).

**Table 2.** The evaluation criteria for the Mulberry-Dyke and Fish-Pond System ecosystem services.

| Level | Score Range | Definition |
|---|---|---|
| I | 4~5 | High ecosystem services |
| II | 3~4 | Relatively high ecosystem services |
| III | 2~3 | General ecosystem services |
| IV | 1~2 | Relatively low ecosystem services |
| V | 0~1 | Low ecosystem services |

Establishing Evaluation Criteria for Quantitative Factors (Table 3). The data related to mulberry leaf production (D1), mulberry fruit production (D2), conventional fish farming production (D3), and ecological fish farming production (D4) were sourced from the Huzhou Agricultural Science and Technology Development Center's Academician and Expert Workstation. Surveys revealed that the annual production of mulberry leaves in the mulberry gardens was approximately 18.75 t/hm$^2$, with pest and disease damage causing a reduction of about 30% to 50%; the annual production of mulberry fruits ranged from 15.00 to 18.75 t/hm$^2$, with a 30% to 50% decrease due to pests and diseases; the annual production of conventionally farmed fish was about 22.50 t/hm$^2$, while that of ecologically farmed fish was about 11.25 t/hm$^2$. In aquaculture, reductions due to climate and fish diseases account for about 20% to 30% and, in the worst-case scenario, all fish may die. The evaluation criteria were established based on the survey results. Indicators D5 to D8 were developed with reference to the relevant literature [52–55]. The Air Quality Index (D9) was based on the "Environmental Air Quality Standards" GB3095-2012 of China [56].

Qualitative data were characterized by five levels of satisfaction (very familiar, quite familiar, moderately familiar, slightly familiar, unfamiliar) (very satisfied, fairly satisfied, generally satisfied, not very satisfied, not satisfied) to represent the membership degree of evaluation indicators to the evaluation factor domain V (Table 4).

**Table 3.** Quantitative factor assessment criteria.

| Criterion Layer | Index Layer | Factor Layer | Level | | | | |
|---|---|---|---|---|---|---|---|
| | | | 5 | 4 | 3 | 2 | 1 |
| B1 Provisioning ecosystem services | C1 Mulberry land production value | D1 (t/hm$^2$) Mulberry leaf production | ≥18.75 | 13.13~18.75 | 11.25~13.13 | 9.38~11.25 | ≤9.38 |
| | | D2 (t/hm$^2$) Mulberry fruit production | ≥18.75 | 15.00~18.75 | 10.50~15.00 | 9.00~10.50 | ≤9.00 |
| | C2 Fishpond production value | D3 (t/hm$^2$) Conventional fish farming production | ≥22.50 | 16.88~22.50 | 11.25~16.88 | 5.63~11.25 | ≤5.63 |
| | | D4 (t/hm$^2$) Ecological fish farming production | ≥11.25 | 8.44~11.25 | 5.63~8.44 | 2.81~5.63 | ≤2.81 |
| B2 Regulating ecosystem services | C3 Basal environment regulation value | D5 (kg/hm$^2$) Fertilizer application intensity | <200.00 | 200.00~250.00 | 250.00~350.00 | 350.00~450.00 | >450.00 |
| | | D6 (kg/hm$^2$) Pesticide application intensity | <2.50 | 2.50~3.00 | 3.00~4.00 | 4.00~4.50 | >4.50 |
| | C4 Climate regulation value | D7 (%) Relative humidity adjustment range | >4.00 | 3.00~4.00 | 2.00~3.00 | 1.00~2.00 | <1.00 |
| | | D8 (°C) Average temperature regulation | >4.00 | 3.00~4.00 | 2.00~3.00 | 1.00~2.00 | <1.00 |
| | | D9 Air quality index | <50.00 | 50.00~100.00 | 100.00~150.00 | 150.00~200.00 | >200.00 |

**Table 4.** Qualitative factor definitions.

| Criterion Layer | Index Layer | Factor Layer | Definition |
|---|---|---|---|
| B3 Culture ecosystem services | C5 Aesthetics value | D10 Plant landscape richness | Hierarchical sense of trees, shrubs, and ground cover vegetation; the diversity of species. |
| | | D11 Seasonal changes in the landscape | Seasonal changes in trees, shrubs, and ground cover vegetation, including both woody and herbaceous plants. |
| | | D12 Overall harmony | The overall sense of harmony within the village formed by the cultural landscapes, streets, alleys, architecture, and vegetation within the village. |
| | | D13 Water clarity | The condition of water bodies in the environment. |

**Table 4.** *Cont.*

| Criterion Layer | Index Layer | Factor Layer | Definition |
|---|---|---|---|
| B3<br>Culture ecosystem services | C5<br>Aesthetics value | D14 Leveling and hardening of road surface | Whether the road conditions are in accordance with the environment, including within the village and inside the Mulberry-Dyke and Fish-Pond System, with visual comfort as the criterion. |
| | C6<br>Education value | D15 Fish and mulberry culture education | The educational significance or value brought by fish–mulberry culture and the research-oriented activities centered around it. |
| | | D16 Humanistic tradition | The existing stories of prominent individuals and their spirit and character within the village that possess propagational and educational significance. |
| | | D17 Cultural propaganda and exhibition | The educational significance or value brought by the promotion and display of folk culture. |
| | | D18 Religious culture propaganda | The spiritual connotations brought by the religious culture atmosphere, as well as the level of understanding and acceptance of it. |
| | C7<br>Leisure and entertainment value | D19 Location conditions | The accessibility of the village's geographical location, transportation convenience, and the natural environment. |
| | | D20 Sanitary conditions | Environmental hygiene conditions. |
| | | D21 Tourism infrastructure | Basic infrastructure including toilets, signage, parking spaces, medical service facilities, etc. |
| | | D22 Agricultural diversity experience | Diverse experiences provided by agricultural products such as freshwater fish, mulberry leaf tea, fruits, rice, sesame oil, etc., based on the raw materials produced in Digang Village, which are either processed or directly sold. |
| | | D23 Experience of the diversity of tourism products | The satisfaction of diversified tourism needs by visiting Digang Village. |
| | | D24 Visual and psychological perception | Experiences of visual and psychological sensations brought about by exploring Digang Village. |
| | C8<br>Cultural heritage value | D25 Village architectural style | Whether the architectural style within the village conforms to the characteristics of the Jiangnan water town and rural farming. |
| | | D26 Village traditional customs | Whether traditional folk customs within the village are fully preserved, whether the atmosphere of folk customs is good, and whether they have distinctive features. |
| | | D27 Ancient bridges and other historical and cultural features | The distinctiveness of cultural landscapes such as ancient bridges, celebrity memorial halls, and the scenic beauty of Nantiao. |
| | | D28 Food culture characteristics | The distinctiveness of Di Gang cuisine, exemplified by the Chen family's dishes and local snacks. |
| | | D29 Fish and mulberry culture characteristics | The distinctive features of the fish–mulberry culture. |

(2)    Determination of weight values

We applied the AHP method to determine the weights of each index. Pairwise comparisons were made among indicators at the same level to construct a judgment matrix. The normalization of weight values and weight vectors was then accomplished through relevant calculation formulas. The weight vector set was obtained through the verification steps [57,58].

(3)    Constructing the affiliation matrix

We determined the membership degree of the evaluated object to the evaluation factor domain and obtained the fuzzy relationship matrix. In this study, the membership degree of quantitative data to the evaluation factor domain was determined through a trapezoidal function, and the membership degree of qualitative data to the evaluation factor domain was obtained through statistical analysis [59].

(4)    Obtaining a composite score for the overall goal

Building on the aforementioned analysis, the overall goal composite score (F) was calculated using the weighted average method, expressed as F = W·R, where W represents the weight vector set and R represents the membership matrix.

### 2.2.3. Data

Data related to the Mulberry-Dyke and Fish-Pond System for the year 2022 were collected through interviews, questionnaires, field surveys, and applications. The study conducted an assessment of the ecosystem services of the Mulberry-Dyke and Fish-Pond System for the year 2022. The quantitative indicators were primarily sourced from the Huzhou Agricultural Science and Technology Development Center's Academician and Expert Workstation, as shown in Table 5. This data encompassed mulberry and fish production as well as the quantities of fertilizers and pesticides used. Air quality index data were obtained from the weather network (http://www.weather.com.cn/ (accessed on 23 February 2023)). Qualitative indicators were gathered through a one-to-one questionnaire survey conducted with tourists and villagers engaged in activities at Digang Village Mulberry-Dyke and Fish-Pond System. A total of 109 questionnaires were initially collected, with 100 valid questionnaires selected after screening to exclude those completed in less than 2 min.

**Table 5.** Quantitative data acquisition.

| Name | Unit | Data | Data Sources |
|---|---|---|---|
| Mulberry land production | t/hm$^2$ | 75.00 | |
| Mulberry fruit production | t/hm$^2$ | 75.00 | |
| Conventional fish farming production | t/hm$^2$ | 90.00 | Huzhou Agricultural Science and |
| Ecological fish farming production | t/hm$^2$ | 45.00 | Technology Development Center's |
| Fertilizer consumption | t | 4.00 | Academician and Expert Workstation |
| Pesticide consumption | CNY | 34,500.00 | |
| Change in relative humidity (July–September 2022) | % | −1.42 | Internal level meteorological information of |
| Change in average temperature (July–September 2022) | °C | −0.42 | Huzhou Municipal Meteorological Bureau |
| AQI | / | 64.90 | http://www.weather.com.cn/ |

## 3. Results

### 3.1. AHP Weight for Each Indicator

Five experts and scholars specializing in ecosystem services research and the planning and design of Digang Village were invited to provide judgments and weights. All tables passed the consistency test (see Appendix A) and the results are presented in Table 6.

**Table 6.** Ecosystem service weight table.

| Criterion Layer | Normalized Weights | Indicator Layer | C-Layer Weight | Normalized Weights | Factor Layer | D-Layer Weight | Normalized Weights | Rank |
|---|---|---|---|---|---|---|---|---|
| B1 | 0.2572 | C1 | 0.3556 | 0.0878 | D1 | 0.2869 | 0.0262 | 14 |
| | | | | | D2 | 0.7131 | 0.0652 | 3 |
| | | C2 | 0.6444 | 0.1693 | D3 | 0.3033 | 0.0503 | 6 |
| | | | | | D4 | 0.6967 | 0.1155 | 1 |
| B2 | 0.2338 | C3 | 0.3786 | 0.0884 | D5 | 0.5000 | 0.0442 | 9 |
| | | | | | D6 | 0.5000 | 0.0442 | 8 |
| | | C4 | 0.6214 | 0.1454 | D7 | 0.2574 | 0.0374 | 10 |
| | | | | | D8 | 0.2568 | 0.0373 | 11 |
| | | | | | D9 | 0.4859 | 0.0706 | 2 |
| | | C5 | 0.1616 | 0.0838 | D10 | 0.2478 | 0.0204 | 22 |
| | | | | | D11 | 0.1330 | 0.0109 | 28 |
| | | | | | D12 | 0.2309 | 0.0190 | 23 |
| | | | | | D13 | 0.1662 | 0.0137 | 26 |
| | | | | | D14 | 0.2221 | 0.0183 | 24 |
| | | | | | D15 | 0.4771 | 0.0567 | 4 |
| | | C6 | 0.2335 | 0.1226 | D16 | 0.2952 | 0.0351 | 12 |
| | | | | | D17 | 0.1512 | 0.0180 | 25 |
| | | | | | D18 | 0.0766 | 0.0091 | 29 |
| B3 | 0.5091 | | | | D19 | 0.1606 | 0.0208 | 20 |
| | | | | | D20 | 0.1753 | 0.0228 | 17 |
| | | C7 | 0.2549 | 0.1320 | D21 | 0.1979 | 0.0257 | 15 |
| | | | | | D22 | 0.1314 | 0.0171 | 26 |
| | | | | | D23 | 0.1679 | 0.0218 | 18 |
| | | | | | D24 | 0.1668 | 0.0217 | 19 |
| | | | | | D25 | 0.1500 | 0.0267 | 13 |
| | | | | | D26 | 0.1402 | 0.0250 | 16 |
| | | C8 | 0.3500 | 0.1706 | D27 | 0.1155 | 0.0206 | 21 |
| | | | | | D28 | 0.3146 | 0.0561 | 5 |
| | | | | | D29 | 0.2798 | 0.0498 | 7 |

At the guideline level (Table 6), it is evident that CESs (B3) play a crucial role in influencing the overall ecosystem services of the Mulberry-Dyke and Fish-Pond System, commanding a weight share of 0.51. Within CESs (B3), the value of aesthetics, education, leisure and entertainment, and cultural heritage across four aspects contributed to this weight. Notably, the value of cultural heritage (C8) emerged as a significant influencing factor for CESs (B3).

In the index layer (Table 6), the cultural heritage value (C8) held the highest weight at 0.17, underscoring its significance as the primary manifestation of the ecosystem services of the Mulberry-Dyke and Fish-Pond System. Notably, cultural heritage value (C8) was most profoundly influenced by factors such as food culture characteristics (D28) and fish and mulberry cultural characteristics (D29). Following closely, the fishpond production value (C1) was of considerable importance, with a weight share of 0.17. Its significance was primarily influenced by ecological fish farming production (D4). Additionally, climate regulation value (C4), leisure and entertainment value (C7), and educational value (C6) exhibited comparable importance, with weight shares of 0.15, 0.13, and 0.12, respectively. Conversely, the remaining indicators—basal environmental regulation value (C3), mulberry land production value (C1), and aesthetic value (C5)—had weight shares of less than 0.0900. This suggests that the value contributed by the ecosystem services of the Mulberry-Dyke and Fish-Pond System predominantly revolves around five aspects: cultural heritage, fishpond production, climate regulation, leisure and entertainment, and education. These indicators exert the most significant influence on the ecosystem services of the Mulberry-Dyke and Fish-Pond System, thus warranting particular attention.

In the factor layer (Table 6), eight indicators—ecological fish farming production (D4), air quality index (D9), mulberry fruit production (D2), fish and mulberry culture

education (D15), food culture characteristics (D28), conventional fish farming production (D3), fish and mulberry culture characteristics (D29), and fertilizer application intensity (D5)—collectively accounted for a weight share of 0.51. Among these, ecological fish farming production (D4) carried the highest weight at 0.1155, underscoring its paramount importance. This implies that these eight indicators were pivotal factors influencing the evaluation, and the ecosystem services of the Mulberry-Dyke and Fish-Pond System emphasize production capacity and content related to fish and mulberry culture.

### 3.2. Fuzzy Comprehensive Evaluation and Results

After obtaining the weights of the indicators in the evaluation system of the Mulberry-Dyke and Fish-Pond ecosystem services, the membership degree matrix is constructed based on the established evaluation factor domain, evaluation criteria, quantitative and qualitative data, and the results are normalized as shown in Tables 7 and 8. Refer to Appendix B for the calculation process. According to the principle of the maximum affiliation function of the FCE method, the affiliation degrees of the ecosystem services of the Mulberry-Dyke and Fish-Pond System were as follows: high level: 0.44, relatively high level: 0.32, general level: 0.10, relatively low level: 0.03, and low level: 0.11. The weighted average result was calculated as [0.44, 0.32, 0.10, 0.03, 0.11] × [5, 4, 3, 2, 1] = 3.97, indicating that the ecosystem services of the Mulberry-Dyke and Fish-Pond System belong to the relatively high level. We multiplied the membership degrees of each criterion, index, and factor by the normalized weights of each layer, as shown in Figure 2.

**Table 7.** Matrix of quantitative factor memberships.

| Factor Layer | Membership Matrix | | | | |
|---|---|---|---|---|---|
| | **5** | **4** | **3** | **2** | **1** |
| D1 Mulberry leaf production | 1.0000 | 0.0000 | 0.0000 | 0.0000 | 0.0000 |
| D2 Mulberry fruit production | 1.0000 | 0.0000 | 0.0000 | 0.0000 | 0.0000 |
| D3 Conventional fish farming production | 1.0000 | 0.0000 | 0.0000 | 0.0000 | 0.0000 |
| D4 Ecological fish farming production | 1.0000 | 0.0000 | 0.0000 | 0.0000 | 0.0000 |
| D5 Fertilizer application intensity | 0.6970 | 0.3030 | 0.0000 | 0.0000 | 0.0000 |
| D6 Pesticide application intensity | 0.0000 | 0.0000 | 0.0000 | 0.0000 | 1.0000 |
| D7 Relative humidity adjustment range | 0.0000 | 0.0000 | 0.0000 | 0.0000 | 1.0000 |
| D8 Average temperature regulation | 0.0000 | 0.0000 | 0.0000 | 0.4200 | 0.5800 |
| D9 Air quality index | 0.0000 | 0.7020 | 0.2980 | 0.0000 | 0.0000 |

**Table 8.** Matrix of qualitative factor memberships.

| Factor Layer | Membership Matrix | | | | |
|---|---|---|---|---|---|
| | **5** | **4** | **3** | **2** | **1** |
| D10 Plant landscape richness | 0.3600 | 0.5900 | 0.0500 | 0.0000 | 0.0000 |
| D11 Seasonal changes in the landscape | 0.3200 | 0.5900 | 0.0800 | 0.0000 | 0.0100 |
| D12 Overall harmony | 0.3100 | 0.5700 | 0.1000 | 0.0200 | 0.0000 |
| D13 Water clarity | 0.1100 | 0.5000 | 0.2900 | 0.0900 | 0.0100 |
| D14 Leveling and hardening of road surface | 0.2800 | 0.5100 | 0.1800 | 0.0300 | 0.0000 |
| D15 Fish and mulberry culture education | 0.3200 | 0.4700 | 0.1600 | 0.0400 | 0.0100 |
| D16 Humanistic tradition | 0.3500 | 0.5000 | 0.1100 | 0.0400 | 0.0000 |
| D17 Cultural propaganda and exhibition | 0.3000 | 0.4400 | 0.2200 | 0.0400 | 0.0000 |
| D18 Religious culture propaganda | 0.1200 | 0.2500 | 0.2500 | 0.3100 | 0.0700 |
| D19 Location conditions | 0.2800 | 0.4700 | 0.2200 | 0.0300 | 0.0000 |
| D20 Sanitary conditions | 0.3300 | 0.4900 | 0.1500 | 0.0200 | 0.0100 |
| D21 Tourism infrastructure | 0.2500 | 0.4900 | 0.2400 | 0.0200 | 0.0000 |
| D22 Agricultural diversity experience | 0.2100 | 0.6100 | 0.1800 | 0.0000 | 0.0000 |

**Table 8.** *Cont.*

| Factor Layer | Membership Matrix | | | | |
| --- | --- | --- | --- | --- | --- |
| | 5 | 4 | 3 | 2 | 1 |
| D23 Experience of the diversity of tourism products | 0.1800 | 0.5900 | 0.2000 | 0.0300 | 0.0000 |
| D24 Visual and psychological perception | 0.3500 | 0.5300 | 0.0900 | 0.0300 | 0.0000 |
| D25 Village architectural style | 0.3700 | 0.4600 | 0.1600 | 0.0100 | 0.0000 |
| D26 Village traditional customs | 0.3500 | 0.4900 | 0.1500 | 0.0100 | 0.0000 |
| D27 Ancient bridges and other historical and cultural features | 0.4200 | 0.4500 | 0.1200 | 0.0100 | 0.0000 |
| D28 Food culture characteristics | 0.3000 | 0.5100 | 0.1700 | 0.0200 | 0.0000 |
| D29 Fish and mulberry culture characteristics | 0.3000 | 0.5500 | 0.1400 | 0.0100 | 0.0000 |

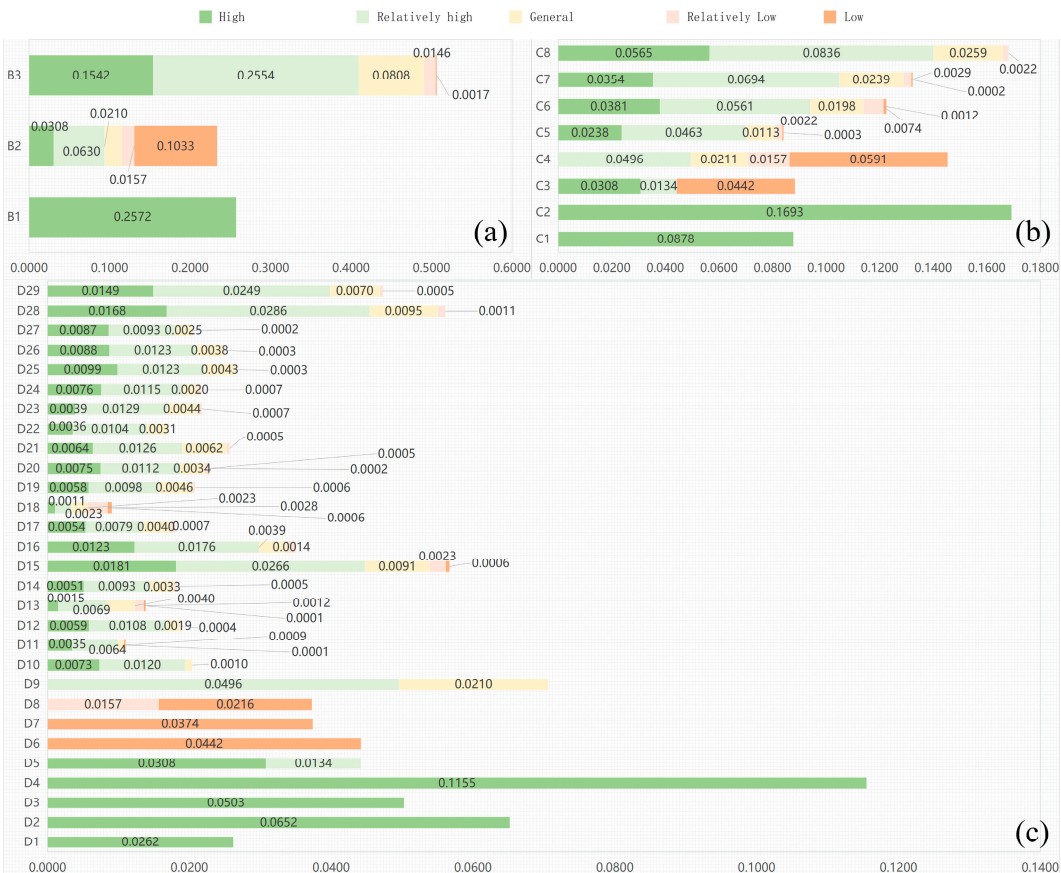

**Figure 2.** Assessment results for each layer. The values shown in the figure are the affiliation functions for each criterion, indicator, and factor layer. (**a**) Affiliation functions for each criterion layer; (**b**) Affiliation functions for each indicator layer; (**c**) Affiliation functions for each factor layer.

The evaluation results of the indicators at the normative level are presented in Figure 2a. Both PESs (B1) and CESs (B3) exhibited high performance, predominantly at the high and relatively high levels. In contrast, RESs (B2) demonstrated a comparatively lower performance, mainly at the relatively low and low levels.

The evaluation results at the indicator and factor levels are depicted in Figure 2b,c. Notably, the four indicators related to the production value of mulberry land (C1) and fishponds (C2) excelled, predominantly at the high level. This suggests that the production capacity of mulberry leaves, mulberry fruits, black carp, and eco-fish in the Mulberry-Dyke and Fish-Pond System is robust. On the other hand, the basal environmental regulation value (C3) and climate regulation value (C4) exhibited lower performance, largely at the relatively low and low levels. This was primarily influenced by factors such as the intensity

of pesticide application (D6), relative humidity regulation (D7), and average temperature regulation (D8). These results indicate deficiencies in pesticide use and the regulation of humidity and temperature in the Mulberry-Dyke and Fish-Pond System. In contrast, the evaluation results for the four values of aesthetic value (C5), educational value (C6), leisure and entertainment value (C7), and cultural heritage value (C8) demonstrated a high degree of affiliation to high and relatively high levels, signifying high performance in these aspects. Among the five main values of ecosystem services in the Mulberry-Dyke and Fish-Pond System, the climate regulation value (C4) was at the low level, indicating an important aspect that requires improvement. However, the top eight indicators in the factor hierarchy largely performed at the high or relatively high levels, suggesting a positive role in promoting the ecosystem services of the Mulberry-Dyke and Fish-Pond System.

## 4. Discussion

### 4.1. An Optimized Management Strategy for the Mulberry-Dyke and Fish-Pond System Based on the AHP-FCE Approach

In our assessment of the Mulberry-Dyke and Fish-Pond System's ecosystem services, we discovered that the RESs (B2) did not achieve high or moderately high levels. Furthermore, the distribution of membership degrees for these services revealed a pronounced bipolar trend, seemingly linked to the interplay among indicators D5 to D9. Specifically, indicators D5 and D6 reflected the growth conditions of mulberry trees to a certain extent, while the area and condition of the mulberry land directly influenced indicators D7, D8, and D9. Consequently, the outcomes for D5, D7, and D8 suggested deficiencies in the management of mulberry lands, underlining the necessity for enhanced professional support in this domain. Reduced and judicious use of pesticides could significantly bolster the regulating service capability of the system, thereby elevating the air quality index. On another front, CESs (B3) emerged as a significant determinant of the system's ecosystem services. This underscores a growing public demand for recreational and cultural facets of the Mulberry-Dyke and Fish-Pond System. Notably, correlations between C5 and C7, as well as C6 and C8, were observed. Thus, reinforcing the key influencers in these aspects is expected to amplify the system's cultural service capacity. Strategic measures include diversifying plant arrangements, augmenting the overall cohesion of the system, enhancing tourism infrastructure, fortifying comprehensive environmental management, and intensively exploring the cultural and historical essence of fish and mulberry practices, along with promoting their unique gastronomical and cultural attributes. Moreover, the factor layer for B3 was predominantly concentrated at a higher level, with a minority at high and average levels, potentially reflecting the evaluators' ambiguity and unclear understanding of related issues. Therefore, contrasting high- and average-level numerical differences could be more indicative, possibly unveiling divergences in opinions about indicator quality. Hence, indicators such as D13, D18, D19, D21, D22, and D23 merit close attention.

When exploring ecosystem services in multifunctional landscapes such as Sankey's fishponds, we must consider the multifunctionality between services and their trade-offs. According to the existing literature [60–62], ecosystem services are not always mutually reinforcing but may be mutually constraining in some cases. Therefore, when planning, constructing, and managing the Mulberry-Dyke and Fish-Pond System, we should not simply seek to maximize a single service but need to consider the balance among services in an integrated manner. The current assessment suggests that there may be some trade-offs between regulating services and provisioning and cultural services. In particular, it is foreseen that the continued enhancement of cultural services will contribute to the diversification of the landscape of the Mulberry-Dyke and Fish-Pond System, the enrichment of recreational facilities, and the improvement of public spaces. However, such changes may also lead to over-intervention in the environment of the Mulberry-Dyke and Fish-Pond System and, consequently, management neglect in the maintenance of vegetation and water bodies, thus affecting the effectiveness of regulating services. In the future management of

the Mulberry-Dyke and Fish-Pond System, we need to weigh the relationship between its ecological benefits and management costs among services.

### 4.2. The Effectiveness of the AHP-FCE Method for Ecosystem Service Assessment in the Mulberry-Dyke and Fish-Pond System

Relative to existing modeling approaches like InVEST and IMAGE renowned for their efficacy in large-scale ecosystem service evaluations, these technologies adopt a modular design and scenario-based data input. They are adept at simulating and forecasting various future possibilities, furnishing quantitative outcomes for stakeholders balancing multiple ecosystem services [63,64]. Nonetheless, these models demand high data quality and volume, making them less suitable for addressing uncertainties, fuzziness, and data gaps in small-scale ecosystem service studies. Contrarily, the AHP-FCE method emerges as a more apt solution for such challenges, producing results that are more comprehensible and interpretable, thereby fostering societal engagement and the inclusion of stakeholder perspectives [65]. This is particularly evident in evaluations of ecosystems where cultural services are prominent as the AHP-FCE method effectively mirrors public sentiments on the value of cultural services. Our findings underscore the pivotal role of cultural services within the ecosystem services of the Mulberry-Dyke and Fish-Pond System, concurrently highlighting managerial issues, resonating with prior research [14,66,67]. The increasing importance of the Mulberry-Dyke and Fish-Pond System's conservation and management is underscored by climate changes and land use transformations in Nanxun town [66]. The AHP-FCE method facilitates the provision of a holistic evaluation indicator system and diagnostic techniques for the future progression of the system's ecosystem services. Moreover, this approach presents a viable resolution to the ambiguities encountered in ecosystem service assessments.

### 4.3. Future Improvement Directions of the AHP-FCE Method for Ecosystem Service Valuations of the Mulberry-Dyke and Fish-Pond System

This evaluation system still offers opportunities for refinement, and future improvements are suggested for enhancing the assessment of ecosystem services in the Mulberry-Dyke and Fish-Pond System. Proposed directions for future enhancement include adjusting indicators based on local planning and development reports and scientific research findings and selectively adding or reducing specific indicators to enhance the scientific rigor of the evaluation system. The monitoring system can be upgraded by accounting for the ratio of base ponds, tallying base crops, and subcategorizing aquatic crops cultivated in the fish ponds. Additionally, the inclusion of quantitative indicators, such as monitoring soil and water quality, can enhance the overall robustness of the evaluation system. It is recommended that the temporal scope of data collection be extended beyond the year 2022 to provide a more comprehensive understanding of the development status of the Mulberry-Dyke and Fish-Pond System. Long-term research efforts will be crucial for gaining insights into the dynamic changes and trends within these ecosystems.

### 5. Conclusions

This study undertook a thorough evaluation of the ecosystem services of the Mulberry-Dyke and Fish-Pond System in Digang Village, Huzhou, utilizing the Analytical Hierarchy Process (AHP) and Fuzzy Comprehensive Evaluation (FCE). Our findings illuminate the paramount role of cultural services within the ecosystem services of the system, particularly emphasizing the substantial impact of cultural heritage values. In contrast, the underperformance of regulating services unveils gaps in mulberry land management and upkeep. These outcomes underscore the necessity to prioritize cultural heritage conservation and enhance regulating services in future management strategies.

This study not only proposes a scientifically robust method for assessing the ecosystem services of the Mulberry-Dyke and Fish-Pond System but also lays a solid foundation for decision-makers to devise more informed and rational management strategies. Additionally, by revealing the significance of cultural services and identifying the areas needing

improvement in regulating services, this research charts new pathways for the sustainable development and ecological protection of the Mulberry-Dyke and Fish-Pond System. These findings hold substantial relevance for fostering the sustainable evolution of agricultural ecosystems in economic, social, and environmental dimensions.

**Author Contributions:** Conceptualization, S.T.; Methodology, Y.L.; Software, M.Z.; Validation, Z.L. and M.Z.; Investigation, S.T.; Resources, M.Z.; Writing—original draft, S.T.; Writing—review & editing, S.T.; Visualization, Z.L. All authors have read and agreed to the published version of the manuscript.

**Funding:** This research received no external funding.

**Data Availability Statement:** No new data were created or analyzed in this study. Data sharing is not applicable to this article.

**Conflicts of Interest:** The authors declare no conflict of interest.

**Appendix A**

**Table A1.** A-B judgment matrix and weights.

| Ecosystem Service Assessment of the Mulberry-Dyke and Fish-Pond System | Provisioning Ecosystem Services B1 | Regulating Ecosystem Services B2 | Culture Ecosystem Services B3 | Normalized Weights |
|---|---|---|---|---|
| Provisioning Ecosystem Services B1 | 1 | 1 | 1/2 | 0.2247 |
| Regulating Ecosystem Services B2 | 1 | 1 | 1/5 | 0.1655 |
| Culture Ecosystem Services B3 | 2 | 5 | 1 | 0.6098 |
| $\lambda_{max} = 3.0940$ | CI = 0.0470 | RI = 0.5200 | | $\sum = 1$ |
| CR = 0.9040 | Satisfying the consistency test | | | |

**Table A2.** Weight table of index layer in Provisioning Ecosystem Services B1.

| Provisioning Ecosystem Services B1 | Mulberry Land Production Value C1 | Fishpond Production Value C2 | Normalized Weights |
|---|---|---|---|
| Mulberry land production value C1 | 1 | 1 | 0.5000 |
| Fishpond production value C2 | 1 | 1 | 0.5000 |
| $\lambda_{max} = 2.0000$ | CI = 0.0000 | RI = 0.0000 | $\sum = 1$ |
| CR = 0.0000 | Satisfying the consistency test | | |

**Table A3.** Factor layer weights in Provisioning Ecosystem Services B1.

| Criterion Layer | Index Layer | Factor Layer | Factor Layer Weight Coefficient |
|---|---|---|---|
| Provisioning ecosystem services B1 | Mulberry land production value C1 | Mulberry leaf production D1 | 0.5000 |
| | | Mulberry fruit production D2 | 0.5000 |
| | Fishpond production value C2 | Conventional fish farming production D3 | 0.5000 |
| | | Ecological fish farming production D4 | 0.5000 |

**Appendix B**

$$C1\text{Mulberry land production value} = w_i \cdot R\text{Mulberry land production value}$$
$$= (0.2869, 0.7131) \begin{vmatrix} 1.0000 & 0.0000 & 0.0000 & 0.0000 & 0.0000 \\ 1.0000 & 0.0000 & 0.0000 & 0.0000 & 0.0000 \end{vmatrix} \tag{A1}$$
$$= (1.0000, 0.0000, 0.0000, 0.0000, 0.0000)$$

$$C2\text{Fishpond production value} = w_i \cdot R\text{Fishpond production value}$$
$$= (0.3033, 0.6967) \begin{vmatrix} 1.0000 & 0.0000 & 0.0000 & 0.0000 & 0.0000 \\ 1.0000 & 0.0000 & 0.0000 & 0.0000 & 0.0000 \end{vmatrix} \tag{A2}$$
$$= (1.0000, 0.0000, 0.0000, 0.0000, 0.0000)$$

$$C3\text{Basal environment regulation value} = w_i \cdot R\text{Basal environment regulation value}$$
$$= (0.5000, 0.5000) \begin{vmatrix} 0.6970 & 0.3030 & 0.0000 & 0.0000 & 0.0000 \\ 0.0000 & 0.0000 & 0.0000 & 0.0000 & 1.0000 \end{vmatrix} \tag{A3}$$
$$= (0.3485, 0.1515, 0.0000, 0.0000, 0.5000)$$

$$C4\text{Climate regulation value} = w_i \cdot R\text{Climate regulation value}$$
$$= (0.2574, 0.2568, 0.4859) \begin{vmatrix} 0.0000 & 0.0000 & 0.0000 & 0.0000 & 1.0000 \\ 0.0000 & 0.0000 & 0.0000 & 0.4200 & 0.5800 \\ 0.0000 & 0.7000 & 0.3000 & 0.0000 & 0.0000 \end{vmatrix} \tag{A4}$$
$$= (0.0000, 0.3411, 0.1448, 0.1078, 0.4063)$$

$$C5\text{Aesthetics value} = w_i \cdot R\text{Aesthetics value}$$
$$= \begin{pmatrix} 0.2478 \\ 0.1330 \\ 0.2309 \\ 0.1664 \\ 0.2221 \end{pmatrix}^T \begin{vmatrix} 0.3600 & 0.5900 & 0.0500 & 0.0000 & 0.0000 \\ 0.3200 & 0.5900 & 0.0800 & 0.0000 & 0.0100 \\ 0.3100 & 0.5700 & 0.1000 & 0.0200 & 0.0000 \\ 0.1100 & 0.5000 & 0.2900 & 0.0900 & 0.0100 \\ 0.2800 & 0.5100 & 0.1800 & 0.0300 & 0.0000 \end{vmatrix} \tag{A5}$$
$$= (0.2838, 0.5526, 0.1343, 0.0262, 0.0030)$$

$$C6\text{Education value} = w_i \cdot R\text{Education value}$$
$$= \begin{pmatrix} 0.4771 \\ 0.2952 \\ 0.1512 \\ 0.7656 \end{pmatrix}^T \begin{vmatrix} 0.3200 & 0.4700 & 0.1600 & 0.0400 & 0.0100 \\ 0.3500 & 0.5000 & 0.1100 & 0.0400 & 0.0000 \\ 0.3000 & 0.4400 & 0.2200 & 0.0400 & 0.0000 \\ 0.1200 & 0.2500 & 0.2500 & 0.3100 & 0.0700 \end{vmatrix} \tag{A6}$$
$$= (0.3105, 0.4575, 0.1612, 0.0607, 0.0101)$$

$$C7\text{Leisure and entertainment value} = w_i \cdot R\text{Leisure and entertainment value}$$
$$= \begin{pmatrix} 0.1606 \\ 0.1753 \\ 0.1979 \\ 0.1314 \\ 0.1679 \\ 0.1668 \end{pmatrix}^T \begin{vmatrix} 0.2800 & 0.4700 & 0.2200 & 0.0300 & 0.0000 \\ 0.3300 & 0.4900 & 0.1500 & 0.0200 & 0.0100 \\ 0.2500 & 0.4900 & 0.2400 & 0.0200 & 0.0000 \\ 0.2100 & 0.6100 & 0.1800 & 0.0000 & 0.0000 \\ 0.1800 & 0.5900 & 0.2000 & 0.0300 & 0.0000 \\ 0.3500 & 0.5300 & 0.1900 & 0.0300 & 0.0000 \end{vmatrix} \tag{A7}$$
$$= (0.2685, 0.5260, 0.1814, 0.0223, 0.0018)$$

$$C8\text{Cultural heritage value} = w_i \cdot R\text{Cultural heritage value}$$
$$= \begin{pmatrix} 0.1500 \\ 0.1402 \\ 0.1155 \\ 0.3146 \\ 0.2798 \end{pmatrix}^T \begin{vmatrix} 0.3700 & 0.4600 & 0.1600 & 0.0100 & 0.0000 \\ 0.3500 & 0.4900 & 0.1500 & 0.0100 & 0.0000 \\ 0.4200 & 0.4500 & 0.1200 & 0.0100 & 0.0000 \\ 0.3000 & 0.5100 & 0.1700 & 0.0200 & 0.0000 \\ 0.3000 & 0.5500 & 0.1400 & 0.0100 & 0.0000 \end{vmatrix} \tag{A8}$$
$$= (0.3314, 0.5040, 0.1515, 0.0131, 0.0000)$$

$$B1\text{Provisioning ecosystem services} = w_i \cdot R\text{Provisioning ecosystem services}$$
$$= (0.3556, 0.6444) \begin{vmatrix} 1.0000 & 0.0000 & 0.0000 & 0.0000 & 0.0000 \\ 1.0000 & 0.0000 & 0.0000 & 0.0000 & 0.0000 \end{vmatrix}$$
$$= (1.0000, 0.0000, 0.0000, 0.0000, 0.0000) \tag{A9}$$

$$B2\text{Regulating ecosystem services} = w_i \cdot R\text{Regulating ecosystem services}$$
$$= (0.3786, 0.6214) \begin{vmatrix} 0.3485 & 0.1515 & 0.0000 & 0.0000 & 0.5000 \\ 0.0000 & 0.3411 & 0.1448 & 0.1078 & 0.4063 \end{vmatrix}$$
$$= (0.1319, 0.2693, 0.0900, 0.0670, 0.4418) \tag{A10}$$

$$B3\text{Culture ecosystem services} = w_i \cdot R\text{Culture ecosystem services}$$
$$= \begin{pmatrix} 0.1616 \\ 0.2335 \\ 0.2549 \\ 0.3500 \end{pmatrix}^{T} \begin{vmatrix} 0.2838 & 0.5526 & 0.1343 & 0.0262 & 0.0030 \\ 0.3105 & 0.4575 & 0.1612 & 0.0607 & 0.0101 \\ 0.2685 & 0.5260 & 0.1814 & 0.0223 & 0.0018 \\ 0.3314 & 0.5040 & 0.1515 & 0.0131 & 0.0000 \end{vmatrix}$$
$$= (0.3028, 0.5066, 0.1586, 0.0287, 0.0033) \tag{A11}$$

$$A\text{Ecosystem service assessment of mulberry} - \text{based fishponds}$$
$$= w_i \cdot R\text{Ecosystem service assessment of mulberry} - \text{base fishponds}$$
$$= (0.5091, 0.2338, 0.2572) \begin{vmatrix} 0.3028 & 0.5066 & 0.1586 & 0.0287 & 0.0033 \\ 0.1319 & 0.2693 & 0.0900 & 0.0670 & 0.4418 \\ 1.0000 & 0.0000 & 0.0000 & 0.0000 & 0.0000 \end{vmatrix}$$
$$= (0.4422, 0.3209, 0.1018, 0.0303, 0.1050) \tag{A12}$$

## Appendix C

Survey Questionnaire

This questionnaire is anonymous, and all information will only be used for this research and not for other purposes. Completing this questionnaire will take approximately 10–15 min. Thank you for your time and effort in supporting scientific research!

[Part One] Basic Information

We would like to know a bit about you. All information is anonymously filled in, and no one will know which answers belong to you.

1. What is your gender? (single choice question) [mandatory question]

Male

Female

2. What is your age? (single choice question) [mandatory question]

Under 18

18–25 years old

26–30 years old

31–40 years old

41–50 years old

51–60 years old

Over 60 years old

3. At the moment of filling out this questionnaire, what is your status? [single choice question] [mandatory question]

Local resident

Tourist

Government

Investor/Entrepreneur (investor, local entrepreneur, local worker)

Expert/Scholar (including university experts, planners, designers, public organization personnel)

[Part Two] Survey on the Mulberry-Dyke and Fish-Pond System in Diggang Village

In this section, 5 points represent being very familiar or very satisfied, 4 points indicate quite familiar or fairly satisfied, 3 points mean moderately familiar or generally satisfied, 2 points denote slightly familiar or not very satisfied, and 1 point signifies being unfamiliar or not satisfied.

(1) Aesthetic Value

1. How do you rate the richness of the plant landscape of the Mulberry-Dyke and Fish-Pond System in Diggang Village?

5, 4, 3, 2, 1

Note: Hierarchical sense of trees, shrubs, and ground cover vegetation; diversity of species.

2. How do you evaluate the seasonal changes in the landscape of the Mulberry-Dyke and Fish-Pond System in Diggang Village?

5, 4, 3, 2, 1

Note: Seasonal changes in trees, shrubs, and ground cover vegetation, including both woody and herbaceous plants.

3. How do you evaluate the overall harmony of the Mulberry-Dyke and Fish-Pond System in Diggang Village?

5, 4, 3, 2, 1

Note: The overall sense of harmony within the village, formed by the cultural landscapes, streets, alleys, architecture, and vegetation within the village.

4. How do you evaluate the clarity of the water bodies in the Mulberry-Dyke and Fish-Pond System in Diggang Village?

5, 4, 3, 2, 1

Note: The condition of water bodies in the environment.

5. How do you evaluate the road surface evenness and hardening in the Mulberry-Dyke and Fish-Pond System in Diggang Village?

5, 4, 3, 2, 1

Note: Whether the road conditions are in accordance with the environment, including within the village and inside the Mulberry-Dyke and Fish-Pond System, with visual comfort as the criterion.

(2) Educational Value

6. How do you evaluate the educational value of the fish and mulberry culture in the context of the Mulberry-Dyke and Fish-Pond System?

5, 4, 3, 2, 1

Note: The educational significance or value brought by fish-mulberry culture and the research-oriented activities centered around it.

7. How do you evaluate the traditional human culture in the context of the Mulberry-Dyke and Fish-Pond System?

5, 4, 3, 2, 1

Note: The existing stories of prominent individuals, their spirit, and character within the village that possess propagational and educational significance.

8. How do you evaluate the cultural promotion and performances related to the Mulberry-Dyke and Fish-Pond System?

5, 4, 3, 2, 1

Note: The educational significance or value brought by the promotion and display of folk culture.

9. How do you evaluate the religious culture in the context of the Mulberry-Dyke and Fish-Pond System?

5, 4, 3, 2, 1

Note: The spiritual connotations brought by the religious culture atmosphere, as well as the level of understanding and acceptance of it.

(3) Recreational Value

10. How do you evaluate the location conditions of the Mulberry-Dyke and Fish-Pond System in Diggang Village?

5, 4, 3, 2, 1

Note: The accessibility of the village's geographical location, transportation convenience, and natural environment.

11. How do you evaluate the hygiene conditions of the Mulberry-Dyke and Fish-Pond System in Diggang Village?

5, 4, 3, 2, 1

Note: Environmental hygiene conditions.

12. How do you evaluate the infrastructure of the Mulberry-Dyke and Fish-Pond System in Diggang Village?

5, 4, 3, 2, 1

Note: Basic infrastructure including toilets, signage, parking spaces, medical service facilities, etc.

13. How do you evaluate the diversity of agricultural products in the Mulberry-Dyke and Fish-Pond System in Diggang Village?

5, 4, 3, 2, 1

Note: Diverse experiences provided by agricultural products such as freshwater fish, mulberry leaf tea, fruits, rice, sesame oil, etc., based on the raw materials produced in Digang Village, which are either processed or directly sold.

14. How do you evaluate the diversity of tourism service products in the Mulberry-Dyke and Fish-Pond System in Diggang Village?

5, 4, 3, 2, 1

Note: Satisfy diversified tourism needs by visiting Digang Village.

15. How do you evaluate your visual and psychological experiences in the Mulberry-Dyke and Fish-Pond System in Diggang Village?

5, 4, 3, 2, 1

Note: Experiences of visual and psychological sensations brought about by exploring Digang Village.

(4) Cultural Heritage Value

16. How do you evaluate the village architectural style in the context of the Mulberry-Dyke and Fish-Pond System?

5, 4, 3, 2, 1

Note: Whether the architectural style within the village conforms to the characteristics of the Jiangnan water town and rural farming.

17. How do you evaluate the village traditional folk customs in the context of the Mulberry-Dyke and Fish-Pond System?

5, 4, 3, 2, 1

Note: Whether traditional folk customs within the village are fully preserved, whether the atmosphere of folk customs is good, and whether they have distinctive features.

18. How do you evaluate the cultural features of ancient buildings (ancient bridges, historical buildings) in the context of the Mulberry-Dyke and Fish-Pond System?

5, 4, 3, 2, 1

Note: The distinctiveness of cultural landscapes such as ancient bridges, celebrity memorial halls, and the scenic beauty of Nantiao.

19. How do you evaluate the food culture in the context of the Mulberry-Dyke and Fish-Pond System?

5, 4, 3, 2, 1

Note: The distinctiveness of Di Gang cuisine, exemplified by the Chen family's dishes and local snacks.

20. How do you evaluate the unique cultural characteristics of the fish and mulberry culture in the context of the Mulberry-Dyke and Fish-Pond System?

5, 4, 3, 2, 1

Note: The distinctive features of the fish-mulberry culture.

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
