# Peer review of "Enhancing Sustainability through Ecosystem Services Evaluation: A Case Study of the Mulberry-Dyke and Fish-Pond System in Digang Village"

_sustainability, doi:10.3390/su16051875_

Round 1

Reviewer 1 Report

Comments and Suggestions for Authors

This manuscript evaluated three types of ecosystem services of a Mulberry-Dyke and Fish-Pond System, Digang Village in China. Digang Village is a world-famous example of traditional and agroecological ecosystems. Scientific evaluation of the ecosystem services of such a system would contribute to a better understanding of its social-economical and ecological values. However, the present form seems not to be well organized, partly due to the lack of novelty and the writing. I have some comments for the authors.

Abstract

1. ecological environment: What did you mean when using such a term here? This term is not a formal scientific term, which has no clear definition.

2. This study only assessed three types of ecosystem services. According to the well-accepted framework proposed by MA, there was a lack of supporting service.

3. Throughout the abstract, it is hard to find the novelty of this study.

Introduction

Lines 30-31: It is not true, at least in the mountain area of southern China, where there is little mechanization and facility implementation.

Line 36: ecological balance. This term is a term out of fashion. Usually, we use ecological resilience or ecological stability to describe the status of an ecosystem to respond to environmental changes.

Lines 48-49: Although human activities dominate this system, it could also support biodiversity maintenance to a certain extent. The delivery of the other three types of ecosystem services depends on supporting services. Thus, to comprehensively evaluate ecosystem services in this unique system, it is not wise to neglect supporting services.

Lines 59-69: This part did not comprehensively review existing studies on MDFPS and propose the novelty compared to previous studies.

Lines 71-73: From the background information, I did not know how many such agriculture systems exist in China and how important Digang Village is compared to other systems.

Lines 76-78: Why did you use the two methods? Please verify the effectiveness of the two methods by citing relevant publications.

Lines 80-82: There was no clear and solid scientific question.

Materials and methods

Lines 86-104: There was no citation in this part.

Figure 1: This map does not present the location of Huzhou in China.

Lines 143-145: Repeated previous paragraph.

Lines 149-150: I am questioning the validation of such criteria since there is no citation here.

Table 3-4: Those tables should be cited in the main text.

Lines 157-159: The data source is too vague. For example, there was no time information for mulberry leaf production. Did you only use data from last year or average data from the latest ten years?

Lines 189-191: Please generally present the main aims and questions of the questionnaire, or attach it as an appendix.

Comments on the Quality of English Language

The English deserved to be refined substantially. 

Author Response

Dear reviewer,

The comments of the reviewer are very helpful and constructive for our research work. The submitted manuscript is the revised version of sustainability-2818343 “Enhancing Sustainability through Ecosystem Services Evaluation a Case Study of the Mulberry-Dyke and Fish-Pond System in Digang Village”. We have elaborately revised the manuscript according to reviewer' reasonable comments and following are the response details.

Regards,

Shuyang Tang, Ziwei Liu, Yumei Li, Mingqin Zhou

==========

Abstract

Point 1: ecological environment: What did you mean when using such a term here? This term is not a formal scientific term, which has no clear definition.

Response 1: Thank you for your suggestion. Your reminder has made us aware of the issue, and as a result, we have changed the phrasing at the beginning of the abstract section to:

"The Mulberry-Dyke and Fish-Pond System, as a paradigm of traditional Chinese agricultural recycling models, represents a distinct ecosystem."

Point 2: This study only assessed three types of ecosystem services. According to the well-accepted framework proposed by MA, there was a lack of supporting service.

Response 2: Thank you for your suggestion. According to the theory of the MA, there indeed exists a category of supporting services. However, supporting services form the basis for the other three types of services and are essentially equivalent to ecological processes. Calculating supporting services would lead to redundancy in calculations. Therefore, our study focuses on the other three types of services. We have explained this in the instruction section and updated the citations.

  1. Vermaat, J.E.; Palt, M.; Piffady, J.; Putnins, A.; Kail, J. The effect of riparian woodland cover on ecosystem service delivery by river floodplains: a scenario assessment. ECOSPHERE 2021, 12, e03716, doi:doi.org/10.1002/ecs2.3716.
  2. Assessment, M.E. Ecosystems and human well-being; Island Press, Washington, DC: 2005; Volume 5.

Point 3: Throughout the abstract, it is hard to find the novelty of this study.

Response 3: Thank you for your suggestion. Regarding the novelty of our research, we offer the following explanation: The core conservation area of the Huzhou Mulberry-Dyke and Fish-Pond System is located in Digang Village, which houses the world's only Mulberry-Dyke and Fish-Pond System Visitor Center and the Huzhou Agricultural Science and Technology Development Center's Academician and Expert Workstation. With the increasing attention from local governments to the Mulberry-Dyke and Fish-Pond System in recent years, conducting an ecosystem service assessment for such GIAHS has become particularly important. Given the small and targeted scale of the research, obtaining relevant data is very challenging. To assess local ecosystem services on a limited data basis, we explored a flexible methodology that is capable of handling data with significant ambiguity and uncertainty.

Here is our revised abstract:

"The Mulberry-Dyke and Fish-Pond System, as a paradigm of traditional Chinese agricultural recycling models, represents a distinct ecosystem. This study focuses on the Mulberry-Dyke and Fish-Pond System in Digang Village, Huzhou, as a typical case. The village serves as a core conservation base for the Huzhou Mulberry-Dyke and Fish-Pond System, hosting the Huzhou Agricultural Science and Technology Development Center's Academician and Expert Workstation and the world's only Mulberry-Dyke and Fish-Pond System Visitor Center. These facilities provide strategic guidance for the conservation, development, planning, and inheritance of the Huzhou Mulberry-Dyke and Fish-Pond System. Considering the unique environment and limitations in data acquisition, this study employed the Analytic Hierarchy Process (AHP) and Fuzzy Comprehensive Evaluation (FCE) to develop an ecosystem service assessment framework encompassing 8 aspects and 29 factors, assessing the provisioning, regulating, and cultural services of the Mulberry-Dyke and Fish-Pond System. The results indicate that the ecosystem services of the Digang Village Mulberry-Dyke and Fish-Pond System perform at a high level, with cultural services playing a significant role in the overall ecosystem services. The regulating services are relatively weak, highlighting deficiencies in mulberry land management, while the capacity of provisioning services is strong. These findings are crucial for understanding the value of ecosystem services in Digang Village's Mulberry-Dyke and Fish-Pond System, identifying management shortcomings, and providing direction for future assessments and management. This study also offers a practical and effective assessment method for ecosystem service evaluation at smaller scales, where the targeted approach and the presence of significant ambiguity and uncertainty in data are prominent. "

Introduction

Point Lines 30-31: It is not true, at least in the mountain area of southern China, where there is little mechanization and facility implementation.

Response 4: Thank you for your suggestion. We recognize that there were issues with our wording, and they have been changed.

"With the advancement of technology, agricultural mechanization and facility-based practices have become the dominant trend in the majority of current agricultural models. "

Point Line 36: ecological balance. This term is a term out of fashion. Usually, we use ecological resilience or ecological stability to describe the status of an ecosystem to respond to environmental changes.

Response 5: Thank you for your suggestion. We have recognized the issues in our phrasing and have made the following change:

"With the advancement of technology, agricultural mechanization and the use of facilities have become the dominant trend in the vast majority of current agricultural models."

Point Lines 48-49: Although human activities dominate this system, it could also support biodiversity maintenance to a certain extent. The delivery of the other three types of ecosystem services depends on supporting services. Thus, to comprehensively evaluate ecosystem services in this unique system, it is not wise to neglect supporting services.

Response 6: Thank you for your suggestion. Regarding supporting services, an explanation has already been provided above. We will add further clarification on this aspect in the article.

Point Lines 59-69: This part did not comprehensively review existing studies on MDFPS and propose the novelty compared to previous studies.

Response 7: Thank you for your suggestion. Research related to the Mulberry-Dyke and Fish-Pond System is more prevalent within China. We have included a review of existing research within China and highlighted the innovative aspects of our study.

Point Lines 71-73: From the background information, I did not know how many such agriculture systems exist in China and how important Digang Village is compared to other systems.

Response 8: Thank you for your suggestion. We have added information about the number of GIAHS within China in the "Study Area" section. However, this study focuses on the Mulberry-Dyke and Fish-Pond System and does not compare the significance among different GIAHS. Additionally, we have made appropriate modifications to the "Study Area" to highlight the representativeness of the research location.

"Currently, China has 19 Globally Important Agricultural Heritage Systems (GIAHS). The Mulberry-Dyke and Fish-Pond System in Huzhou, Zhejiang (located at 37°12′18″N, 120°17′40″E), is situated on the plains south of Lake Taihu, within a subtropical climate zone. The annual average temperature ranges from 17.8°C to 18.2°C, with annual precipitation between 1348 mm and 1723 mm. In 2017, it was recognized as a GIAHS [30]. Digang Village, located in the Nanxun town of Huzhou, serves as an important monitoring site for the Mulberry-Dyke and Fish-Pond heritage system. It is home to the world's only Mulberry-Dyke and Fish-Pond Visitor Center and the Huzhou Agricultural Science and Technology Development Center's Academician and Expert Workstation, which focuses on the conservation and planning of the Mulberry-Dyke and Fish-Pond System. The total area of the village is about 643 hectares, consisting of arable land, gardens, forests, grasslands, water bodies, water conservancy facilities, construction land, and other types of land. The Mulberry-Dyke and Fish-Pond System itself covers approximately 220 hm², with the central village area about 130 hm². The main agricultural activities include freshwater fish farming and sericulture. The most important agricultural species within the system include 2 types of silkworms, 4 types of mulberry trees, 7 types of fish, and 3 types of fruits and vegetables. The village has a registered population of about 1141 households, of which approximately 280 provide heritage tourism services, and the per capita income of farmers operating agricultural heritage is 6500 yuan. The village houses the core conservation area of the Huzhou Mulberry-Dyke and Fish-Pond System, covering about 66 hm², including approximately 23-25 hm² of mulberry gardens and 33-35 hm² of fish ponds. This study focuses on the central village of Digang and the core conservation area within the village."

Point Lines 76-78: Why did you use the two methods? Please verify the effectiveness of the two methods by citing relevant publications.

Response 8: Thank you for your suggestion. These two methods allow for flexible development of evaluation frameworks and quantitative analysis. We will add references in this section to substantiate the effectiveness of the methods.

Point Lines 80-82: There was no clear and solid scientific question.

Response 9: Thank you for your suggestion. We have already provided an explanation of the relevant scientific issues in the above content. Here, we would like to reiterate that our scientific question is: Given the limitations in data availability and methodological application, how can we effectively assess ecosystem services with smaller scales and strong specificity, and explore their contributions to local agricultural sustainability?

The following are our revisions for lines 48-82:

"In the research specifically focusing on the Mulberry-Dyke and Fish-Pond System, most of the literature comes from contributions by Chinese scholars, including investigations into the historical eco-economic context of the Mulberry-Dyke and Fish-Pond System [18,19], studies on ecological restoration strategies [20,21], analyses of landscape patterns [22,23], material cycling models [24], and sustainability capabilities [25]. How-ever, these studies have not yet comprehensively considered the combined benefits of the services provided by the Mulberry-Dyke and Fish-Pond System, especially in terms of the spiritual well-being obtained by humans, thus necessitating further in-depth re-search to unveil their comprehensive performance in these areas.

This study focuses on Digang Village within the core conservation area of the Mulberry-Dyke and Fish-Pond System in Huzhou, Zhejiang Province, China. The village, characterized by the typical features of the Jiangnan water town plains, is surrounded by water on all sides and has developed over thousands of years relying on the Mulberry-Dyke and Fish-Pond System. The aim is to explore an assessment method for specific ecosystem services suitable for smaller research scales under conditions of significant ambiguity and uncertainty in data. Given the current conditions in Digang Village, this study adopts a methodology that combines the Analytical Hierarchy Process (AHP) with Fuzzy Comprehensive Evaluation (FCE) to flexibly construct an assessment framework and perform quantitative analysis [26,27]. Based on the Millennium Ecosystem Assessment (MA) classification system for ecosystem services, and considering that supporting services are essentially equivalent to ecological processes—where calculating them could lead to redundancy [28,29]—the study focuses on quantifying PES, RES, and CES. "

  1. Zhou, Q. During the late Qing Dynasty and the Republic of China era, the Mulberry-Dyke and Fish-Pond System in the Pearl River Delta was closely linked with the ecological and economic environment. JOURNAL OF SOUTH CHINA AGRICULTURAL UNIVERSITU (SOCIAL SCIENCE EDITION) 2013, 12(03): 142-150.
  2. Wu, J. The agricultural practices around basic ponds in Shunde during the Ming, Qing, and Republic of China periods and the economic transformation. Ancient and Modern Agriculture 2011, (01): 96-104.
  3. Nie, C; Luo, S; Zhang, J; Li, H; Zhao, Y. The dike-pond system in the Pearl River Delta: degradation following recent land use alterations and measures for their ecological restoration. ACTA ECOLOGICA SINICA 2003, (09): 1851-1860.
  4. Chen, C; Ye, Y; Huang, G; Gong, Q; Liu, X. Scale effect of the dike pond' s multifunctionality and ecological restoration strategy in the Guangdong-Hong Kong-Macau Greater Bay Area. ACTA ECOLOGICA SINICA 2021, 41(09): 3394-3405, doi: 10.5846 /stxb202004301072.
  5. Wang, X; Xia, L; Deng, S. Spatial-temporal Changes in Dike –pond Land in Nanhai District based on RS and GIS. RESOURCES & INDUSTRIES 2011, 13(04): 55-60. DOI: 10.13776/j.cnki.resourcesindustries.2011.04.002.
  6. Wang, C; Huang, S; Miao, J; Wang X. Characteristics and Driving Factors of Landscape Pattern Evolution of Dike-Ponds in the Pearl River Delta: A Case Study of Shunde District, Foshan. Chinese Landscape Architecture 2022, 38(06): 75-80, doi: 10.19775/j.cla.2022.06.0075.
  7. Chen, Y; Chen, J; Xie, X; Zhang, X; Liu, K; Dai, J; Zhang, C. Research Progress of Nitrogen and Phosphorus Pollution Characteristics, Migration and Transformation in Mulberry-fish Pond Subsystem. Guangdong Agricultural Sciences 2021, 48(11): 74-87, doi: 10.16768/j.issn.1004-874X.2021.11.010.
  8. Astudillo, M.F.; Thalwitz, G.; Vollrath, F. Modern analysis of an ancient integrated farming arrangement: life cycle assessment of a mulberry dyke and pond system. INTERNATIONAL JOURNAL OF LIFE CYCLE ASSESSMENT 2015, 20, 1387-1398, doi: 10.1007/s11367-015-0950-3.
  9. Villacreses, G.; Jijón, D.; Nicolalde, J.F.; Martínez-Gómez, J.; Betancourt, F. Multicriteria Decision Analysis of Suitable Location for Wind and Photovoltaic Power Plants on the Galápagos Islands. Energies 2022, 16, doi: 10.3390/en16010029.
  10. Nicolalde, J.F.; Yaselga, J.; Martínez-Gómez, J. Selection of a Sustainable Structural Beam Material for Rural Housing in Latin América by Multicriteria Decision Methods Means. Applied Sciences 2022, 12, doi: 10.3390/app12031393.
  11. Vermaat, J.E.; Palt, M.; Piffady, J.; Putnins, A.; Kail, J. The effect of riparian woodland cover on ecosystem service delivery by river floodplains: a scenario assessment. ECOSPHERE 2021, 12, e03716, doi: doi.org/10.1002/ecs2.3716.
  12. Assessment, M.E. Ecosystems and human well-being; Island Press, Washington, DC: 2005; Volume 5.

Materials and methods

Point Lines 86-104: There was no citation in this part.

Response 10: Thank you for your suggestion. We have added additional references.

"In 2017, it was recognized as a Globally Important Agricultural Heritage System [30]."

  1. Zhou, R.; Huang, L.; Wang, K.; Hu, W. From Productive Landscape to Agritouristic Landscape? The Evidence of an Agricultural Heritage System—Zhejiang Huzhou Mulberry-Dyke and Fish-Pond System. Land 2023, 12, doi:10.3390/land12051066.

Point Figure 1: This map does not present the location of Huzhou in China.

Response 10: Thank you for your suggestion. We have made adjustments to the figure 1.

Point Lines 143-145: Repeated previous paragraph.

Response 11: Thank you for your suggestion. We apologize for this oversight and have corrected it.

Point Lines 149-150: I am questioning the validation of such criteria since there is no citation here.

Response 12: Thank you for your suggestion. This data was provided by professionals from the " Huzhou Agricultural Science and Technology Development Center's Academician and Expert Workstation," who are primarily responsible for guiding the development of the Mulberry-Dyke and Fish-Pond System. We have standardized this data based on the research results. The results indicate that the annual mulberry leaf production in the Mulberry-Dyke and Fish-Pond System is approximately 1250kg/mu, with a reduction of approximately 30% to 50% due to disease and pest damage. The annual mulberry fruit production is approximately between 1000 to 1250 kg/mu, with a reduction of approximately 30% to 50% due to disease and pest damage. Conventional fish farming production have an annual yield of approximately 1500kg/mu, while Ecological fish farming production have an annual yield of approximately 750kg/mu. In aquaculture, climate and fish diseases account for about 20% to 30% of production reduction, and in the worst-case scenario, all the fish may die. Considering the conversion factor in China where 1 hectare equals 15 mu, we have converted the units to tons per hectare. We have identified some errors in the previous data conversion process, and these have been corrected.

" Establishing Evaluation Criteria for Quantitative Factors (Table 3). The data related to mulberry leaf production (D1), mulberry fruit production (D2), conventional fish farming production (D3), and ecological fish farming production (D4) were sourced from the Huzhou Agricultural Science and Technology Development Center's Academician and Expert Workstation. Surveys revealed that the annual production of mulberry leaves in mulberry gardens is approximately 18.75t/ hm², with pest and dis-ease damage causing a reduction of about 30% to 50%; the annual production of mulberry fruits ranges from 15.00 to 18.75t/ hm², with a 30% to 50% decrease due to pests and diseases; the annual production of conventionally farmed fish is about 22.50t/ hm², while that of ecologically farmed fish is about 11.25t/ hm². In aquaculture, reductions due to climate and fish diseases account for about 20% to 30%, and in the worst-case scenario, all fish may die. The evaluation criteria were established based on the survey results. Indicators D5 to D8 were developed with reference to relevant literature [48-51]. The Air Quality Index (D9) is based on the "Environmental Air Quality Stand-ards" GB3095-2012 of China."

Point Table 3-4: Those tables should be cited in the main text.

Response 12: Thank you for your suggestion. We have made the correction.

Point Lines 157-159: The data source is too vague. For example, there was no time information for mulberry leaf production. Did you only use data from last year or average data from the latest ten years?

Response 13: Thank you for your suggestion. The data sources, as explained in detail above, were obtained through a survey conducted in 2023 with the assistance of the "Huzhou Agricultural Science and Technology Development Center's Academician Expert Workstation." The survey covered relevant data from 2022, and the assessment of the Mulberry-Dyke and Fish-Pond System's ecosystem services also reflects the local conditions as of 2022.

Point Lines 189-191: Please generally present the main aims and questions of the questionnaire, or attach it as an appendix.

Response 14: Thank you for your suggestion. We have uploaded it as an appendix.

Survey Questionnaire

This questionnaire is anonymous, and all information will only be used for this research and not for other purposes. Completing this questionnaire will take approximately 10-15 minutes. Thank you for your time and effort in supporting scientific research!

【Part One】Basic Information

We would like to know a bit about you. All information is anonymously filled in, and no one will know which answers belong to you.

1、What is your gender? (single choice question) [mandatory question]

Male

Female

2、What is your age? (single choice question) [mandatory question]

Under 18

18-25 years old

26-30 years old

31-40 years old

41-50 years old

51-60 years old

Over 60 years old

3、At the moment of filling out this questionnaire, what is your status? [single choice question] [mandatory question]

Local resident

Tourist

Government

Investor/Entrepreneur (investor, local entrepreneur, local worker)

Expert/Scholar (including university experts, planners, designers, public organization personnel)

【Part Two】Survey on the Mulberry-Dyke and Fish-Pond System in Diggang Village

In this section, 5 points represent being very familiar or very satisfied, 4 points indicate quite familiar or fairly satisfied, 3 points mean moderately familiar or generally satisfied, 2 points denote slightly familiar or not very satisfied, and 1 point signifies being unfamiliar or not satisfied.

(1) Aesthetic Value

1、How do you rate the richness of the plant landscape of the Mulberry-Dyke and Fish-Pond System in Diggang Village?

5, 4, 3, 2, 1

Note: Hierarchical sense of trees, shrubs, and ground cover vegetation; diversity of species.

2、How do you evaluate the seasonal changes in the landscape of the Mulberry-Dyke and Fish-Pond System in Diggang Village?

5, 4, 3, 2, 1

Note: Seasonal changes in trees, shrubs, and ground cover vegetation, including both woody and herbaceous plants.

3、How do you evaluate the overall harmony of the Mulberry-Dyke and Fish-Pond System in Diggang Village?

5, 4, 3, 2, 1

Note: The overall sense of harmony within the village, formed by the cultural landscapes, streets, alleys, architecture, and vegetation within the village.

4、How do you evaluate the clarity of the water bodies in the Mulberry-Dyke and Fish-Pond System in Diggang Village?

5, 4, 3, 2, 1

Note: The condition of water bodies in the environment.

5、How do you evaluate the road surface evenness and hardening in the Mulberry-Dyke and Fish-Pond System in Diggang Village?

5, 4, 3, 2, 1

Note: Whether the road conditions are in accordance with the environment, including within the village and inside the Mulberry-Dyke and Fish-Pond System, with visual comfort as the criterion.

(2) Educational Value

6、How do you evaluate the educational value of the fish and mulberry culture in the context of the Mulberry-Dyke and Fish-Pond System?

5, 4, 3, 2, 1

Note: The educational significance or value brought by fish-mulberry culture and the research-oriented activities centered around it.

7、How do you evaluate the traditional human culture in the context of the Mulberry-Dyke and Fish-Pond System?

5, 4, 3, 2, 1

Note: The existing stories of prominent individuals, their spirit, and character within the village that possess propagational and educational significance.

8、How do you evaluate the cultural promotion and performances related to the Mulberry-Dyke and Fish-Pond System?

5, 4, 3, 2, 1

Note: The educational significance or value brought by the promotion and display of folk culture.

9、How do you evaluate the religious culture in the context of the Mulberry-Dyke and Fish-Pond System?

5, 4, 3, 2, 1

Note: The spiritual connotations brought by the religious culture atmosphere, as well as the level of understanding and acceptance of it.

(3) Recreational Value

10、How do you evaluate the location conditions of the Mulberry-Dyke and Fish-Pond System in Diggang Village?

5, 4, 3, 2, 1

Note: The accessibility of the village's geographical location, transportation convenience, and natural environment.

11、How do you evaluate the hygiene conditions of the Mulberry-Dyke and Fish-Pond System in Diggang Village?

5, 4, 3, 2, 1

Note: Environmental hygiene conditions.

12、How do you evaluate the infrastructure of the Mulberry-Dyke and Fish-Pond System in Diggang Village?

5, 4, 3, 2, 1

Note: Basic infrastructure including toilets, signage, parking spaces, medical service facilities, etc.

13、How do you evaluate the diversity of agricultural products in the Mulberry-Dyke and Fish-Pond System in Diggang Village?

5, 4, 3, 2, 1

Note: Diverse experiences provided by agricultural products such as freshwater fish, mulberry leaf tea, fruits, rice, sesame oil, etc., based on the raw materials produced in Digang Village, which are either processed or directly sold.

14 How do you evaluate the diversity of tourism service products in the Mulberry-Dyke and Fish-Pond System in Diggang Village?

5, 4, 3, 2, 1

Note: Satisfy diversified tourism needs by visiting Digang Village.

15、How do you evaluate your visual and psychological experiences in the Mulberry-Dyke and Fish-Pond System in Diggang Village?

5, 4, 3, 2, 1

Note: Experiences of visual and psychological sensations brought about by exploring Digang Village.

(4) Cultural Heritage Value

16、How do you evaluate the village architectural style in the context of the Mulberry-Dyke and Fish-Pond System?

5, 4, 3, 2, 1

Note: Whether the architectural style within the village conforms to the characteristics of the Jiangnan water town and rural farming.

17、How do you evaluate the village traditional folk customs in the context of the Mulberry-Dyke and Fish-Pond System?

5, 4, 3, 2, 1

Note: Whether traditional folk customs within the village are fully preserved, whether the atmosphere of folk customs is good, and whether they have distinctive features.

18、How do you evaluate the cultural features of ancient buildings (ancient bridges, historical buildings) in the context of the Mulberry-Dyke and Fish-Pond System?

5, 4, 3, 2, 1

Note: The distinctiveness of cultural landscapes such as ancient bridges, celebrity memorial halls, and the scenic beauty of Nantiao.

19、How do you evaluate the food culture in the context of the Mulberry-Dyke and Fish-Pond System?

5, 4, 3, 2, 1

Note: The distinctiveness of Di Gang cuisine, exemplified by the Chen family's dishes and local snacks.

20、How do you evaluate the unique cultural characteristics of the fish and mulberry culture in the context of the Mulberry-Dyke and Fish-Pond System?

5, 4, 3, 2, 1

Note: The distinctive features of the fish-mulberry culture.

Reviewer 2 Report

Comments and Suggestions for Authors

The study's objective is to evaluate its integral value concerning agricultural production, ecological environment, and heritage conservation. Employing the Analytical Hierarchy Process (AHP) and Fuzzy Comprehensive Evaluation (FCE) methodologies. However, the methodology is very developed, so the contribution of the research must be made very clear, with respect to what has been published. 

Related to the introduction, in the bibliographic review, the contribution of the research is not clearand more references should be included to clarify the contribution. The authors must try to create a theoretical framework to observe the interest of the article. Include references for AHP methods like

Villacreses, G., Jijón, D., Nicolalde, J. F., Martínez-Gómez, J., & Betancourt, F. (2022). Multicriteria Decision Analysis of Suitable Location for Wind and Photovoltaic Power Plants on the Galápagos Islands. Energies, 16(1), 29. 

Nicolalde, J. F., Yaselga, J., & Martínez-Gómez, J. (2022). Selection of a Sustainable Structural Beam Material for Rural Housing in Latin América by Multicriteria Decision Methods Means. Applied Sciences, 12(3), 1393. 

The methodology is well developed; however, explaining the ranges of values used in Table 3 would be interesting. Evaluation criteria for quantitative factors. 

The results and discussion of results are well developed. However, as the methodology is highly developed, a discussion of results, highlighting the contribution of the article is necessary. 

Reviewer 3 Report

Comments and Suggestions for Authors

It is a very interesting and high-quality manuscript. The authors recognized the importance of the problem and dealt with this topic concisely and systematically. I suggest to the authors small corrections that are not necessary if they think that they will not significantly affect the quality of the paper.

Reviewer 4 Report

Comments and Suggestions for Authors

The article is relevant, informative and good. However, I have some questions:

1. It seems that the word «Digang Village» should be added to the key word;

2. The introduction requires information about the Digang Village, which is considered the subject of the article;

3. It would be more understandable if the introduction presented the current environmental situation in Digang Village;

4. The paragraph in lines 54-70 should be in the section «Research methods»;

5. The abbreviations in this sentence are unclear: “The study established an ecosystem service evaluation system for Mulberry-Dyke 74 and Fish-Pond System based on the PES, RES, and CES categories according to the MA 75 classification of ES" . Lines 74-76. I think this will be incomprehensible to readers;

6. In which program was Figure 2 made? Can you enlarge the figure?

Round 2

Reviewer 1 Report

Comments and Suggestions for Authors

Although the authors revised the manuscript substantially, my main concerns are not well addressed, especially regarding the lack of novelty and neglect of supporting services throughout the text.

Comments on the Quality of English Language

 Moderate editing of the English language is required.

Round 3

Reviewer 1 Report

Comments and Suggestions for Authors

No further comments
